# Association of Dietary Vitamin C Consumption with Serum Klotho Concentrations

**DOI:** 10.3390/foods12234230

**Published:** 2023-11-23

**Authors:** Yan Wang, Mingyang Wu, Lu Xiang, Si Liu, Gang Luo, Qian Lin, Lin Xiao

**Affiliations:** Xiangya School of Public Health, Central South University, Changsha 410078, China; 226911039@csu.edu.cn (Y.W.); mingyangwu2016@163.com (M.W.); xianglu66@csu.edu.cn (L.X.); 226911029@csu.edu.cn (S.L.); luogang@csu.edu.cn (G.L.); linqian@csu.edu.cn (Q.L.)

**Keywords:** dietary vitamin C, serum klotho concentrations, antioxidant, NHANES

## Abstract

Background: Klotho is widely recognized as a protein that combats aging and possesses antioxidative characteristics, which have been implicated in the pathophysiology of numerous diseases. There is emerging evidence suggesting that the consumption of dietary nutrients, particularly those rich in antioxidants, could be associated with serum Klotho concentrations. Dietary vitamin C is one of the critical nutrients that possesses antioxidant properties. Nonetheless, the association between dietary vitamin C consumption and serum Klotho concentrations remains unclear. Objective: Aiming to evaluate the relationship between serum Klotho concentrations and dietary vitamin C consumption among Americans aged 40 to 79, we conducted a population-based study. Methods: From the National Health and Nutrition Examination Survey (NHANES) conducted between 2007 and 2016, a grand total of 11,282 individuals who met the criteria were selected as eligible participants for the study. Serum Klotho concentrations were measured using an ELISA kit that is commercially available. Trained interviewers evaluated the consumption of dietary vitamin C in the diet through a 24-hour dietary recall technique. A generalized linear model was used to evaluate the correlation between the consumption of dietary vitamin C in the diet and serum Klotho concentrations. Further examination was conducted using restricted cubic spline (RCS) analysis to explore the non-linear correlation between dietary vitamin C consumption in the diet and serum Klotho concentrations. Results: After accounting for possible confounding factors, serum Klotho concentrations rose by 1.17% (95% confidence interval (CI): 0.37%, 1.99%) with every standard deviation (SD) rise in dietary vitamin C consumption. With the first quintile of dietary vitamin C consumption as a reference, the percentage change of serum Klotho concentrations in the fifth quintile of dietary vitamin C consumption was 3.66% higher (95% CI: 1.05%, 6.32%). In older, normal-weight, and male participants, the subgroup analysis revealed a stronger correlation between dietary vitamin C consumption and serum Klotho concentrations. Analysis of RCS showed a linear positive association between dietary vitamin C consumption and the levels of serum Klotho concentrations. Conclusion: The findings of this research indicate a strong and positive correlation between dietary vitamin C consumption and serum Klotho concentrations among the general adult population in the United States. Further studies are needed to validate the present findings and to explore specific mechanisms.

## 1. Introduction

Klotho is a multifunctional protein that is widely distributed throughout the body, including but not limited to the brain, kidney, parathyroid, and pituitary glands. Ever since Klotho was first described over two decades ago, substantial efforts have been made to clarify its physiologic functions, of which the anti-aging function is likely the most studied [1]. It has been demonstrated that a single low-dose injection of Klotho improves cognitive function in aged monkeys [2]. Research by Kuro-o et al. suggests that Klotho-deficient mice or mice with a silenced Klotho gene result in accelerated aging and a shortened lifespan, while overexpression of Klotho extends the lifespan of mice [1]. The serum Klotho concentrations have been found to be negatively correlated with age in a number of cross-sectional studies in humans [3,4]. There is a correlation between higher serum Klotho concentration and better cognition, frail psychological components, dependence, and milder falls [5].

Klotho is involved in preventing oxidative stress [6] and inflammation [4], which trigger a number of aging-related diseases. It has been shown that Klotho exerts an anti-aging role by inhibiting the insulin/IGF-1 signaling pathway to induce the expression of superoxide dismutase (SOD), which further contributes to enhancing resistance to oxidative stress and eliminating reactive oxygen species (ROS) in mice [7]. Treatment with serum-soluble Klotho has been proven to alleviate cardiac aging in mice by suppressing excessive oxidative stress, inflammation, and apoptosis [8]. In addition, the supplementation of melatonin with antioxidant properties in mice reduced oxidative load and ameliorated the aging phenotype caused by Klotho deficiency [9]. Furthermore, supplementation with antioxidant substances could increase the expression level of Klotho protein, such as Artemisia argyi water extract (AAW) [10], vitamin E [11], quercetin [12], nicotinamide (NAM) [13], epigallocatechin-3-gallate (EGCG) [14], sulforaphane (SFN) [15], and so on. Importantly, a positive association between the intake of antioxidant substances and Klotho levels has been demonstrated in population studies [16].

Vitamin C, which is also referred to as ascorbic acid, is considered a vital nutrient for the human body [17]. As a natural antioxidant, vitamin C is highly effective in scavenging reactive oxygen species and helps reduce damage caused by free radicals. In a model of renal injury caused by hyperoxaluria, vitamin C supplementation could restore the kidney Klotho protein level reduced by hydroxy-L-proline (HLP)-induced hyperoxaluria [18]. At the same time, both decreased Klotho levels and shortened telomere length can be used as indicators of biological aging [19], and there is a positive correlation between dietary vitamin C consumption and human telomere length [20]. These studies all seem to confirm a direct or indirect association between vitamin C and Klotho, implying that increased dietary vitamin C consumption can increase Klotho levels. However, the direct relationship between dietary vitamin C consumption and serum Klotho concentrations remains unclear.

The Klotho family consists of three proteins (α-Klotho, β-Klotho, and γ-Klotho), which have different organ-specific expression and functions in vivo [21]. Among them, α-Klotho has been the subject of extensive research and is currently the most studied, existing in three types: transmembrane α-Klotho, soluble α-Klotho, and secreted α-Klotho [22]. Abscission α-Klotho and secretory α-Klotho are collectively referred to as soluble α-Klotho (S-Klotho). As the major form of circulating Klotho, S-Klotho can be detected in the blood. A variety of functions can be performed by S-Klotho, like a hormone in cells and tissues that do not express Klotho, including inhibition of oxidative stress and inflammation [23]. ELISA-measured levels of S-Klotho have been established as a straightforward and significant marker of aging [24], thus finding widespread usage in population-based studies to investigate factors influencing aging-related diseases and their associations [16,25,26]. For this study, we sought to complement existing epidemiological research on the relationship between dietary vitamin C consumption and serum Klotho concentrations. To achieve this objective, we analyzed data from the National Health and Nutrition Examination Survey (NHANES), which includes information on S-Klotho (hereafter referred to as Klotho) levels detected during the survey.

## 2. Materials and Methods

### 2.1. Study Population

The National Health and Nutrition Examination Survey (NHANES) is a comprehensive cross-sectional survey conducted by the Centers for Disease Control and Prevention (CDC) and designed to be nationally representative. Its primary objective is to evaluate the health and nutrition status of the entire U.S. population. Previous relevant reports have described the study design and baseline characteristics of NHANES in detail (https://wwwn.cdc.gov/nchs/nhanes/, accessed on 18 March 2023). All research protocols utilized in NHANES have undergone rigorous approval by the National Center for Health Statistics Research Ethics Review Board. Each participant signed a written informed consent form agreeing to participate in the NHANES, including specimen storage and continuing studies. For a more in-depth understanding of NHANES research methods and protocols, a detailed description can be obtained from the official NHANES website.

Participant recruitment is performed by NHANES staff. All data utilized in this study is reliable, and any unreliable data is excluded from analysis. Sample selection for NHANES followed these stages, in order: (1) selection of primary sampling units (PSUs), which are counties or small groups of contiguous counties; (2) selection of segments within PSUs that constitute a block or group of blocks containing a cluster of households; (3) selection of specific households within segments; and (4) selection of individuals within a household. About 12,000 people per 2-year cycle were asked to participate in NHANES. Response rates varied by year. Participants are located in counties across the United States. For the current investigation, a total of 50,588 adults were recruited from the 2007–2016 NHANES study cycle. It is important to acknowledge that the study might be influenced by potential alterations or deviations arising from various physiological and pathological factors. Consequently, individuals with specific characteristics were excluded from the analysis, namely: (1) those without serum Klotho concentration data (*n* = 36,824); (2) those lacking dietary vitamin C consumption data from the initial 24-hour dietary review (*n* = 804); and (3) those without covariate data (*n* = 1678). Following these exclusions, the final cohort comprised 11,282 participants, who were included in the subsequent analyses (see Figure 1).

### 2.2. Determination of Serum Klotho Concentrations

In this study, frozen serum samples obtained from individuals aged 40 to 79 years were utilized to measure serum Klotho concentrations. The analysis of serum Klotho concentrations was carried out using a commercial ELISA kit (IBL International, Japan). Quality assurance measures were implemented by averaging the results of two repeat analyses, and samples with repeat results differing by more than 10% were subjected to reanalysis. The detection sensitivity of the serum Klotho concentrations was determined to be 4.33 pg/mL. To establish a reference range, a set of 114 samples from healthy donors was evaluated, yielding a range of 285.8 to 1638.6 pg/mL with a mean concentration of 698.0 pg/mL. All analysis samples were stored at −80 °C for future use. Subsequently, the samples were transported from the Centers for Disease Control and Prevention to the Northwestern Lipid Metabolism and Diabetes Research Laboratory in the Division of Metabolism, Endocrinology, and Nutrition at the University of Washington for further analysis [27]. The process of testing serum Klotho concentrations was performed by NHANES investigators, and the dataset was acquired from the NHANES website (https://wwwn.cdc.gov/nchs/nhanes/, accessed on 18 March 2023) for analysis in this study.

### 2.3. Assessment of Vitamin C Consumption

All participants in NHANES were eligible to participate in the dietary interview portion. In this study, data were collected from the first 24-hour dietary recall interview, which was conducted by the Mobile Examination Center (MEC). The dataset from the Dietary Data Questionnaire was utilized to record the intake of vitamin C in the diet using the “What We Eat in America” tool, which was created by the Department of Agriculture and the Department of Health and Human Services in the United States. For quality control, interviewers were trained before and tested during the dietary interview. The dataset employed in this study does not encompass the inclusion of dietary supplements. The dietary vitamin C consumption was expressed in mg/d. The data on dietary vitamin C consumption used in this study came from not only vegetables and fruits but also vegetable juice, fruit juice, fruit drinks, and similar items.

### 2.4. Covariate Adjustment

Covariates were selected based on numerous influencing factors listed in previous literature and the correlation with serum Klotho concentrations [28]. Demographic variables include age, income poverty ratio (PIR), sex (male and female), education (no high school diploma, only high school diploma and college or higher), and race (non-Hispanic white, non-Hispanic black, other Hispanic and Mexican Americans, or others). The variable examined is body mass index (BMI), calculated as weight divided by height squared (kg/m^2^). Health-behavior-related variables included serum cotinine, alcohol consumption (<12 drinks per year and ≥12 drinks per year), and dietary energy intake. Health-status-related variables included history of diabetes (yes/no) and history of hypertension (yes/no). The glomerular filtration rate (eGFR, mL/min/1.73 m^2^) was assessed using the chronic kidney disease epidemiological collaborative equation [29].

### 2.5. Methods of Statistical Analysis

In categorical variables, counts (percentages) are used for representation, and chi-square tests are used for comparison. Continuous variables with normal distributions were expressed as mean ± standard deviation, and analysis of variance (ANOVA) was used for comparison between groups. Continuous variables with skewed distributions were presented as the median (25–75th) and compared using the Kruskal–Wallis test. The serum Klotho concentrations were log transformed to better achieve a normal distribution. Dietary vitamin C consumption was analyzed not only as a continuous variable but also as a categorical variable (quintiles), and Q1 was used as a reference. The analysis of multivariate linear regression was conducted to assess the association between dietary vitamin C consumption and serum Klotho concentrations. Using the medians of dietary vitamin C consumption in each quintile as continuous variables, linear trend tests were performed.

Three regression models were used in this study. Model 1 was a crude model with no adjustment for any covariates. Model 2 was a minimally adjusted model, with adjustments for age and sex. Model 3, a fully adjusted model, was further adjusted for health behavior factors and health status based on Model 2, including BMI, PIR, education, ethnicity, serum cotinine, alcohol consumption, diabetes, hypertension, estimates of glomerular filtration rate (eGFR), and dietary energy intake for subsequent subgroup analysis. In order to make the results more intuitive, in the analysis where dietary vitamin C consumption was treated as a continuous variable, we calculated the percentage change in serum Klotho concentrations for each standard deviation (SD) increase of the dietary vitamin C consumption according to the following formula (e^SD×β^ − 1) × 100%, with 95% confidence intervals (CI) calculated as (e^[SD×(β±1.96×SE)]^ − 1) × 100%. In the analysis where dietary vitamin C consumption was treated as a categorical variable, we employed the first quintile as a reference, computed the percentage change in serum Klotho concentrations for the remaining four quintiles, and applied the ensuing formula (e^β^ − 1) × 100%. Additionally, the determination of 95% CI followed the formula (e^(β±1.96×SE)^ − 1) × 100%. β and SE are the regression coefficient and standard error, respectively.

Significant correlations between age, BMI, sex, and serum Klotho concentrations have been reported in previous literature. Subgroup analysis was conducted in this study to further explore the potential effects of different subgroups on serum Klotho concentrations. Statistical interactions were performed using the Wald test. A regression model was constructed with interactions of dietary vitamin C consumption with age, BMI, and gender, and the *p*-values of the interaction terms were analyzed. In addition, the restricted cubic spline (RCS) was utilized to assess the dose-response relationship between dietary vitamin C consumption and serum Klotho concentrations in all participants and subgroups with four knots at the 5th, 35th, 65th, and 95th percentiles.

In order to obtain a nationally representative estimate of the results of the analysis in the United States, the MEC weights recommended by the NHANES database were used. All data collection and statistical analysis were performed using R 4.2.2. A *p* < 0.05 was considered statistically significant (bilateral).

## 3. Results

### 3.1. Basic Characteristics of All Participants

As illustrated in Table 1, baseline characteristics of all participants according to the quartiles of dietary vitamin C consumption are presented. A total of 11,282 participants were enrolled in the study according to the inclusion and exclusion criteria. The mean age of all participants was 57.78 ± 10.81 years. The dietary vitamin C consumption of all participants ranged from 0 to 1617.80 mg/d, and the median serum Klotho concentrations (25th to 75th) were 800.55 (653.95, 990.00) pg/mL. Participants were split into 5544 males (49.10%) and 5738 females (50.90%). There was a statistically significant difference between dietary vitamin C consumption quartiles and all characteristics except alcohol consumption, hypertension, and eGFR. The participants with higher dietary vitamin C consumption were more likely to be of normal BMI, male, non-smokers, non-diabetics, have high dietary energy intakes, and tend to have higher levels of education and income. Additionally, participants with a higher intake of dietary vitamin C had higher serum Klotho concentrations.

### 3.2. Association between Serum Klotho Concentrations and Dietary Vitamin C Consumption

Table 2 shows that dietary vitamin C consumption had a significant positive correlation with serum Klotho concentrations, according to the results of multivariate linear regression analyses (Model 1~3). All participants had an increase in serum Klotho concentration of 1.33% (95% CI: 0.56%, 2.12%, *p* = 0.001) per SD increase in dietary vitamin C consumption in Model 1. In both Model 2 (percent change = 1.54%, 95%CI: 0.78%, 2.31%, *p* < 0.001) and Model 3 (percent change = 1.17%, 95%CI: 0.37%, 1.99%, *p* = 0.006) that had been adjusted for relative confounding factors, the positive correlation remained significant. A stratified analysis of dietary vitamin C consumption and serum Klotho concentrations based on the quantiles of dietary vitamin C consumption was performed in order to determine the exact correlation between dietary vitamin C consumption and serum Klotho concentrations. The variations in serum Klotho concentrations observed across Models 1 to 3 exhibited a positive correlation with higher consumption. Particularly, in comparison to Models 1 and 3, the most substantial incremental rise in serum Klotho concentrations was evident in age- and sex-adjusted Model 2 for individuals with elevated dietary vitamin C consumption. In the linear trend analysis of dietary vitamin C consumption and serum Klotho level, the values of *P* for trend in Model 1~3 all supported the linear relationship between dietary vitamin C consumption and serum Klotho concentrations.

### 3.3. Subgroup Analysis

A subgroup stratification analysis was conducted in this study, as shown in Table 3. After adjusting for potential confounding factors, serum Klotho concentrations and dietary vitamin C consumption were stratified by age, BMI, and sex. Dietary vitamin C consumption and serum Klotho concentrations were significantly correlated in older, normal-weight, and male participants. There was a significant increase in serum Klotho concentrations in older participants (percent change = 1.90%, 95% CI: 0.37%, 3.43%), normal-weight participants (percent change = 2.38%, 95% CI: 1.17%, 3.61%), and male participants (percent change = 1.80%, 95% CI: 0.73%, 2.89%) with the per SD increase in dietary vitamin C consumption. Furthermore, BMI had a significant interaction with dietary vitamin C consumption and serum Klotho concentrations in stratified analyses (*p* for interaction = 0.009). A restricted cubic spline (RCS) was further used to assess the dose-response relationship between dietary vitamin C consumption and serum Klotho concentrations. According to Figure 2 and Appendix A, there was a linear dose-response relationship between dietary vitamin C consumption and serum Klotho concentrations (*p* for nonlinear = 0.510). There was a significant linear association between dietary vitamin C consumption and serum Klotho concentrations in older participants (*p* for nonlinear = 0.330) and male participants (*p* for nonlinear = 0.655). However, there was a significant nonlinear association between dietary vitamin C consumption and serum Klotho concentrations in normal-weight participants (*p* for nonlinearity = 0.099).

## 4. Discussion

The present study aimed to investigate the correlation between dietary vitamin C consumption and serum Klotho concentrations by analyzing a substantial population dataset from NHANES. The findings revealed a noteworthy relationship between dietary vitamin C consumption and serum Klotho concentrations, even after accounting for potential confounding factors. Specifically, a positive dose-effect correlation was observed across all participants. Notably, this correlation exhibited greater prominence in older individuals, normal-weight persons, and male participants.

An increasing number of studies are currently dedicated to exploring the impact of Klotho protein levels on chronic diseases, making it progressively vital to investigate the determinants influencing Klotho protein levels. Population-based studies are gaining momentum in this realm [16,30,31,32]. We observed that vitamin C, with its antioxidant and anti-inflammatory properties, is strongly linked to Klotho protein-related chronic conditions, such as cognitive impairment [33] and cardiovascular disease [34]. However, to our knowledge, only one population-based study has delved into the association between vitamin C and Klotho protein, and its findings are congruent with the results of our present investigation, which demonstrated a positive correlation between heightened vitamin C intake and elevated Klotho protein levels. Nonetheless, the former study did not thoroughly explore the positive relationship. Our study, on the other hand, extensively explored the potential factors contributing to the positive correlation between dietary vitamin C consumption and serum Klotho concentrations within various subgroups of the population. Additionally, we examined whether age, body mass index, and sex interacted in shaping this positive association.

Klotho protein is generally considered to be a major component factor that inhibits aging and is likely to be a new therapeutic target for various anti-aging interventions. The correlation between vitamin C and serum Klotho levels was demonstrated in animal models. In exploring the antioxidant effects of skeels fruit extract (SDE) and compound walnut oil capsules in a mouse model of D-galactose-induced aging, vitamin C was used as a positive control, and the results showed that vitamin C elevated Klotho mRNA expression in the liver and brain [35,36]. Immune-104 is a plant complex composed of vitamin C and other substances, and higher doses of Immune-6 could induce anti-inflammatory Klotho release [37]. Furthermore, it is well known that there is a strong association between Klotho and telomere length. The use of a natural telomere activator containing vitamin C in rats revealed increased telomerase expression in the brain [38]. It has been demonstrated that vitamin C could increase telomerase activity in humans [39]. The aforementioned studies directly or indirectly demonstrated a positive association between vitamin C and Klotho, which is in agreement with the results of the present study. Moreover, our results are consistent with, but extend, another epidemiological finding, which did not explore the positive association in depth [16]. Our study investigated the possible reasons for the positive association between dietary vitamin C consumption and serum Klotho concentrations in different subgroups of the population and examined whether age, BMI, and gender played an interactive role in the positive association.

In addition, it is important to note that vitamin C is a water-soluble vitamin and the body cannot synthesize or store vitamin C on its own due to the loss of a key enzyme in the biosynthetic pathway [40], so we must meet needs by consuming sufficient amounts from outside sources to prevent vitamin C deficiency, including the potentially fatal deficiency scurvy [41]. Typically, the intake of dietary vitamin C is the main determinant of the body’s vitamin C levels [42,43]. Extensive epidemiological investigations have corroborated the widespread occurrence of vitamin C deficiency in the global populace, leading to a heightened incidence of diseases linked to such insufficiency, which can include potentially fatal conditions like scurvy. This deficiency has been linked to various health issues, including cardiovascular disease, congestive heart failure, malignancies, chronic inflammation, metabolic disorders, and non-communicable diseases such as cataracts [33,41,44,45]. The findings from this study substantiate the notion that augmenting the dietary intake of vitamin C may enhance the potential for averting age-related diseases. Consequently, it becomes imperative to ensure adequate vitamin C consumption throughout the entire population, not solely for preventing maladies attributed to vitamin C deficiency but also for mitigating ailments associated with the aging process.

The exact molecular mechanism between dietary vitamin C consumption and serum Klotho concentrations is not fully understood. Combined with previous literature, we speculate that the mechanism between the two may be related to the antioxidant and anti-inflammatory properties possessed by vitamin C. On the one hand, oxidative stress plays an important role in the decrease in Klotho expression [46]. As the most effective antioxidant in plasma, vitamin C can react rapidly with various oxygen radicals and scavenge them effectively, thus achieving antioxidant effects [47]. Mitochondria are the main intracellular sites for the production of endogenous ROS, and vitamin C can reduce the formation of mitochondrial ROS and increase the SOD and glutathione peroxidase (GSH-Px) activities in isolated rat liver mitochondria [48]. SOD and GSH-Px are endogenous antioxidants in the human body and play a synergistic role with the exogenous antioxidant vitamin C from the diet to maintain or re-establish redox homeostasis. Therefore, the antioxidant effect of vitamin C may contribute to the stability of Klotho protein expression. On the other hand, both systemic and local inflammation may reduce the expression level of Klotho protein in the kidney [49]. In animal experiments as well as clinical trials, vitamin C has been shown to alleviate the inflammatory response by modulating inflammatory factor production as well as inhibiting inflammatory mediator infiltration [50]. In addition, the positive regulatory effect of vitamin C by inhibiting oxidative stress levels and thereby increasing Klotho levels has been demonstrated in rats with a hyperoxaluria model [18]. Therefore, we speculate that vitamin C supplementation may help to combat the down-regulation of Klotho protein expression levels caused by inflammation. Nevertheless, the correlation between vitamin C and Klotho, as well as the precise regulatory mechanisms, remain unreported in population-based studies.

Previous studies have shown a decrease in serum Klotho concentrations with increasing age [51]. In this study, to explore the potential age-specific relationship between dietary vitamin C consumption and serum Klotho concentrations, we additionally performed a subgroup analysis. Stratified analysis results showed that increased dietary vitamin C consumption was significantly associated with increased serum Klotho concentrations in older participants but not in middle-aged participants. This may be related to the difference in the compensatory capacity of the antioxidant system between the older and middle-aged participants. After 60 years of age, the compensatory capacity of the antioxidant system of the human body is significantly reduced. Moreover, the lipid metabolism capacity of the elderly population is weak, and it has been proven that an increase in lipids can lead to an increase in systemic inflammation and oxidative stress [52,53]. Vitamin C has the functions of scavenging free radicals, reducing the damage reaction of free radicals, decreasing the content of malondialdehyde (MDA), and significantly increasing the serum SOD content in the human body [54]. In addition, vitamin C can reduce inflammation by inhibiting NF-κB and reducing pro-inflammatory mediators [45]. Therefore, in the older participants, dietary vitamin C consumption may influence the expression of Klotho levels through its antioxidant and anti-inflammatory effects, thus having an impact on the elevation of serum Klotho concentrations in the organism. Although the vitamin C-Klotho relationship was more prominent among elderly adults, the age-vitamin C interaction displayed an insignificantly modified effect on the levels of Klotho. Therefore, the age-specific relationship between dietary vitamin C consumption and serum Klotho concentrations should be viewed with caution [17,55,56].

Previous studies have demonstrated the correlation between BMI and serum Klotho concentrations [57]. Therefore, stratified analyses of dietary vitamin C consumption and serum Klotho concentrations were conducted using BMI as a stratification element. The results showed that increased dietary vitamin C consumption was significantly associated with elevated serum Klotho concentrations in normal-weight participants, but this association was not found in overweight and obese participants. Vitamin C may influence the regulation of lipocalin, which is thought to reduce hepatic lipid accumulation and inflammation [58]. Overweight and obesity can lead to low systemic inflammation and redox imbalances with increased oxidative stress [52,53]. Therefore, we hypothesized that the positive effect of dietary vitamin C consumption on overweight and obese participants has been counteracted. However, further studies are required to clarify the relationship between dietary vitamin C consumption and serum Klotho concentrations.

Our results demonstrated a significant positive association between dietary vitamin C consumption and elevated serum Klotho concentrations in male participants, while no such correlation was observed in female participants. Consistent with the findings of several studies [41,59,60], which indicate lower vitamin C levels in men compared to women, it suggests that the impact of increased vitamin C levels on Klotho concentrations may be more pronounced in males. Nevertheless, the sex-specific differences between dietary vitamin C consumption and serum Klotho concentrations remain uncertain and warrant further investigation in future studies.

The findings of this study revealed that elevating dietary vitamin C consumption exerts a positive influence against certain diseases associated with diminished Klotho protein levels, yielding beneficial effects for the entire population. In specific vulnerable subgroups, such as the elderly, obese, and male populations, who are at a heightened risk of chronic illnesses and possess lower life expectancies compared to other groups, increased dietary vitamin C consumption may prove particularly beneficial in augmenting Klotho protein levels and subsequently mitigating the onset of diseases linked to Klotho protein decline. Conversely, the impact of increased vitamin C intake on disease prevention appears less pronounced in middle-aged, normal-weight, and female populations, indicating the potential need for alternative approaches to minimize disease risk in these demographic groups.

The study utilized data from the NHANES database, employing standardized and processed questionnaire methods and detection methods to collect comprehensive covariate information to facilitate large-sample analyses. The overall study design was nationally representative. Nevertheless, this investigation has some limitations. Firstly, being a cross-sectional study, it precludes the establishment of causal and temporal relationships between dietary vitamin C consumption and serum Klotho concentrations. Secondly, the assessment of dietary vitamin C consumption relied on data from the first day of the 24-hour dietary recall interview, omitting potential factors like seasonal and geographic variations in diet. Thirdly, despite adjusting for potential confounders through a review of prior literature, certain unaccounted confounding variables may persist, such as individual dietary preferences attributed to genetic factors. Fourthly, as the NHANES database only encompasses serum Klotho concentrations for participants aged 40–79 years, the generalizability of results to individuals younger than 40 years or older than 79 years remains uncertain. Fifthly, although the study encompassed a multi-racial population, it was primarily centered within the United States, raising questions about the generalizability of the results to populations of the same race in other countries. Sixth, the analysis in this study relied on dietary vitamin C data obtained from dietary recalls, which may be susceptible to recall bias. Seventh, regrettably, due to the unavailability of the present data, we were unable to select a more compelling serum vitamin C dataset for analysis. Eighth, the assessment of dietary vitamin C consumption may be influenced by factors such as an individual’s nutritional status, intestinal health, nutrient-nutrient interactions, nutrient-drug interactions, and other variables that could affect the evaluation of dietary vitamin C consumption.

## 5. Conclusions

The current study elucidated a positive relationship between serum Klotho concentrations and dietary vitamin C consumption within a population aged 40–79 years in the United States. Substantially stronger associations were observed among older individuals, male participants, and normal-weight persons. Considering the widespread prevalence of vitamin C deficiency worldwide, these results underscore the significant role of dietary vitamin C consumption in the context of anti-aging mechanisms. However, it is imperative to further validate these findings through prospective studies in the future.

## Figures and Tables

**Figure 1 foods-12-04230-f001:**
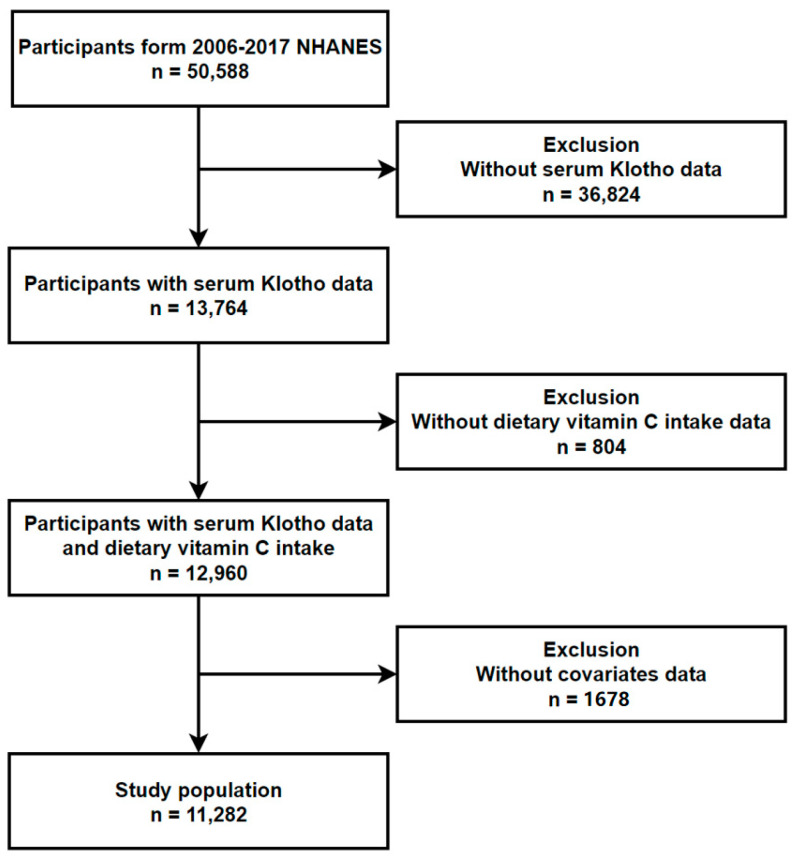
Flowchart of the study population.

**Figure 2 foods-12-04230-f002:**
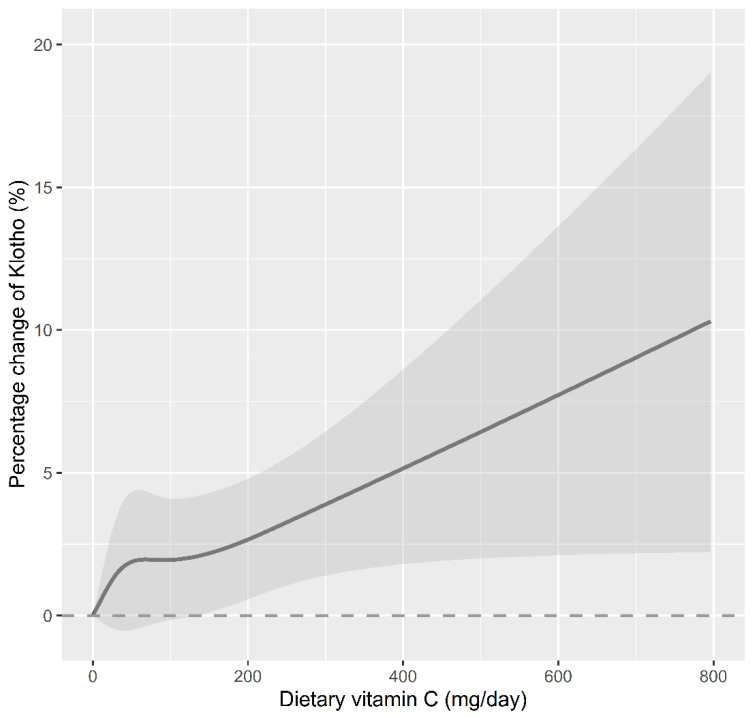
The dose-response relationship between dietary vitamin C consumption and the percent change in Klotho concentration. Point value estimation (solid line) and 95% confidence interval calculation (dashed line) were estimated by a restrictive cubic spline analysis model, knotted at the 5th, 35th, 65th, and 95th percentiles. Age, sex, BMI, PIR, education attainment, ethnicity, serum cotinine, alcohol consumption, diabetes, hypertension, and dietary energy intake were adjusted in the model. *p* for non-linearity is 0.510.

**Table 1 foods-12-04230-t001:** Based characteristics of all participants (*n* = 11,282).

Characteristic	All	The Quintile of Dietary Vitamin C Intake (mg/day)					
		Quintile 1	Quintile 2	Quintile 3	Quintile 4	Quintile 5	*p* Value
Number of participants	11,282	2250	2262	2254	2259	2257	
Dietary vitamin C intake, mean (SD)	81.00 (90.78)	8.24 (5.06)	26.40 (5.62)	52.58 (9.73)	97.31 (16.67)	220.31 (109.64)	**<0.001**
Serum Klotho concentration, pg/mL, median (25th–75th)	800.55 (653.95, 990.00)	787.50 (643.50, 973.72)	797.85 (644.58, 989.98)	802.30 (660.22, 994.20)	794.20 (647.10, 981.90)	821.70 (671.00, 1003.90)	**0.001**
Age, years, mean (SD)	57.78 (10.81)	56.52 (10.77)	56.70 (10.59)	58.24 (10.95)	58.70 (10.90)	57.75 (10.74)	**<0.001**
Sex, *n* (%)							**<0.001**
male	5544 (49.10)	1101 (48.9)	1063 (47.0)	1055 (46.8)	1083 (47.9)	1242 (55.0)	
female	5738 (50.90)	1149 (51.1)	1199 (53.0)	1199 (53.2)	1176 (52.1)	1015 (45.0)	
BMI, kg/m^2^, mean (SD)	29.89 (6.74)	30.35 (7.30)	30.01 (6.62)	29.95 (6.74)	29.66 (6.51)	29.46 (6.47)	**<0.001**
Race/Ethnicity, *n* (%)							**<0.001**
Non-Hispanic White	5203 (46.10)	1084 (48.2)	1107 (48.9)	1052 (46.7)	1051 (46.5)	909 (40.3)	
Non-Hispanic Black	2234 (19.80)	502 (22.3)	423 (18.7)	394 (17.5)	425 (18.8)	490 (21.7)	
Other Hispanic	1199 (10.60)	224 (10.0)	225 (9.9)	249 (11.0)	228 (10.1)	273 (12.1)	
Mexican American or Other	2646 (23.5)	440 (19.6)	507 (22.4)	559 (24.8)	555 (24.6)	585 (25.9)	
Educational attainment, *n* (%)							**<0.001**
<High school	2950 (26.10)	745 (33.1)	607 (26.8)	583 (25.9)	527 (23.3)	488 (21.6)	
High school	2519 (22.30)	556 (24.7)	582 (25.7)	480 (21.3)	476 (21.1)	425 (18.8)	
College or above	5813 (51.50)	949 (42.2)	1073 (47.4)	1191 (52.8)	1256 (55.6)	1344 (59.5)	
PIR, mean (SD)	2.65 (1.65)	2.26 (1.54)	2.56 (1.62)	2.76 (1.66)	2.85 (1.67)	2.84 (1.69)	**<0.001**
Serum cotinine, ng/mL, median (25–75th)	0.04 (0.01, 2.54)	0.13 (0.02, 187.75)	0.05 (0.01, 64.12)	0.03 (0.01, 0.36)	0.03 (0.01, 0.23)	0.03 (0.01, 0.20)	**<0.001**
Alcohol consumption, *n* (%)							0.138
≥12 drinks/year	3199 (28.40)	625 (27.8)	655 (29.0)	667 (29.6)	655 (29.0)	597 (26.5)	
<12 drinks/year	8083 (71.60)	1625 (72.2)	1607 (71.0)	1587 (70.4)	1604 (71.0)	1660 (73.5)	
Diabetes, *n* (%)							**<0.001**
No	8569 (76.00)	1709 (76.0)	1652 (73.0)	1667 (74.0)	1740 (77.0)	1801 (79.8)	
Yes	2713 (24.00)	541 (24.0)	610 (27.0)	587 (26.0)	519 (23.0)	456 (20.2)	
Hypertension, *n* (%)							0.230
No	5133 (45.50)	989 (44.0)	1010 (44.7)	1027 (45.6)	1044 (46.2)	1063 (47.1)	
Yes	6149 (54.50)	1261 (56.0)	1252 (55.3)	1227 (54.4)	1215 (53.8)	1194 (52.9)	
eGFR, mL/min/1.73 m^2^, mean (SD)	84.00 (19.64)	83.94 (20.69)	83.62 (20.10)	84.26 (19.11)	83.20 (19.77)	84.96 (18.45)	0.034
Dietary energy intake, kcal/day, mean (SD)	2023.66 (914.05)	1747.34 (848.38)	1941.31 (836.79)	2045.95 (906.66)	2072.94 (901.60)	2310.06 (977.78)	**<0.001**

Note: SD: standard deviation; BMI: body mass index; PIR: family poverty income ratio; eGFR: estimated glomerular filtration rate. The quintile of dietary vitamin C consumption ranges: Quintile 1: 0 to 17.10 mg/day; Quintile 2: 17.11 to 36.80 mg/day; Quintile 3: 36.81 to 70.60 mg/day; Quintile 4: 70.61 to 129.78 mg/day; and Quintile 5: 129.79 to 1617.80 mg/day. Bold indicates significance with *p* < 0.05.

**Table 2 foods-12-04230-t002:** The association between dietary vitamin C consumption and serum Klotho concentrations among all participants.

Dietary Vitamin C Consumption (mg/day)	Percent Changes (%) and 95% CI					
	Model 1	*p* Value	Model 2	*p* Value	Model 3	*p* Value
Per SD increases	1.33 (0.56, 2.12)	**0.001**	1.54 (0.78, 2.31)	**<0.001**	1.17 (0.37, 1.99)	**0.006**
Quintile 1	Ref.		Ref.		Ref.	
Quintile 2	0.69 (−1.75, 3.19)	0.585	0.97 (−1.43, 3.44)	0.434	0.77 (−1.60, 3.20)	0.530
Quintile 3	0.68 (−1.71, 3.14)	0.581	1.18 (−1.22, 3.63)	0.342	0.51 (−1.68, 2.74)	0.653
Quintile 4	0.48 (−1.54, 2.54)	0.644	1.09 (−0.88, 3.10)	2.833	0.37 (−1.69, 2.48)	0.726
Quintile 5	4.08 (1.51, 6.72)	**0.002**	4.89 (2.38, 7.47)	**<0.001**	3.66 (1.05, 6.32)	**0.007**
*p* for trend		**<0.001**		**<0.001**		**0.011**

Note: CI: confidence interval; SD: standard deviation. Model 1 was a crude model; Model 2 was adjusted for age and sex; and Model 3 was further adjusted for BMI, PIR, ethnicity, education attainment, serum cotinine, alcohol consumption, diabetes or not, hypertension or not, eGFR, and dietary energy intake. The quintile of dietary vitamin C consumption ranges: Quintile 1: 0 to 17.10 mg/day; Quintile 2: 17.11 to 36.80 mg/day; Quintile 3: 36.81 to 70.60 mg/day; Quintile 4: 70.61 to 129.78 mg/day; and Quintile 5: 129.79 to 1617.80 mg/day. Bold indicates significance with *p* < 0.05.

**Table 3 foods-12-04230-t003:** The relationship between dietary vitamin C consumption and Klotho concentrations, stratified by age, BMI, and sex.

Participants	Dietary Vitamin C Consumption (mg/day)	Percent Changes (%) and 95% CI	*p* Value	* *p* for Interaction
Age subgroup				0.457
Age < 60 years	Per SD increases	0.69 (−0.23, 1.62)	0.147	
	Quintile 1	Ref.		
	Quintile 2	2.03 (−1.15, 5.31)	0.219	
	Quintile 3	0.14 (−3.09, 3.49)	0.933	
	Quintile 4	−0.10 (−3.11, 3.00)	0.948	
	Quintile 5	3.51 (0.10, 7.03)	**0.048**	
	*p* for trend		**0.087**	
Age ≥ 60 years	Per SD increases	1.90 (0.37, 3.43)	**0.017**	
	Quintile 1	Ref.		
	Quintile 2	−1.53 (−5.20, 2.28)	0.43	
	Quintile 3	0.94 (−2.82, 4.84)	0.63	
	Quintile 4	0.87 (−2.68, 4.54)	0.64	
	Quintile 5	3.38 (−1.08, 8.05)	0.14	
	*p* for trend		0.0720	
BMI subgroup				**0.009**
BMI < 25 kg/m^2^	Per SD increases	2.38 (1.17, 3.61)	**<0.001**	
	Quintile 1	Ref.		
	Quintile 2	4.10 (−1.85, 10.22)	0.189	
	Quintile 3	−0.93 (−6.39,4.86)	0.749	
	Quintile 4	4.12 (−0.81, 9.30)	0.108	
	Quintile 5	8.27 (2.98, 13.84)	**0.003**	
	*p* for trend		**0.002**	
BMI ≥ 25 kg/m^2^	Per SD increases	0.62 (−0.38, 1.63)	0.231	
	Quintile 1	Ref.		
	Quintile 2	−0.13 (−2.86, 2.69)	0.939	
	Quintile 3	0.98 (−1.70, 3.73)	0.481	
	Quintile 4	−0.93 (−3.62, 1.73)	0.490	
	Quintile 5	2.13 (−1.05, 5.40)	0.196	
	*p* for trend		0.239	
Sex subgroup				0.146
Male	Per SD increases	1.80 (0.73, 2.89)	**0.002**	
	Quintile 1	Ref.		
	Quintile 2	1.85 (−1.31, 5.10)	0.259	
	Quintile 3	1.85 (−1.47, 5.28)	0.283	
	Quintile 4	0.34 (−2.84, 3.62)	0.838	
	Quintile 5	5.68 (1.97, 9.51)	**0.004**	
	*p* for trend		**0.009**	
Female	Per SD increases	0.42 (−0.73, 1.59)	0.480	
	Quintile 1	Ref.		
	Quintile 2	−0.09 (−3.52, 3.47)	0.962	
	Quintile 3	−0.83 (−3.73, 2.17)	0.587	
	Quintile 4	0.37 (−2.75, 3.60)	0.817	
	Quintile 5	1.66 (−1.82, 5.26)	0.358	
	*p* for trend		0.260	

Note: CI: confidence interval; SD: standard deviation; BMI: body mass index. The quintile of dietary vitamin C consumption ranges: Quintile 1: 0 to 17.10 mg/day; Quintile 2: 17.11 to 36.80 mg/day; Quintile 3: 36.81 to 70.60 mg/day; Quintile 4: 70.61 to 129.78 mg/day; and Quintile 5: 129.79 to 1617.80 mg/day. * *p* value for the interaction of vitamin C intake with age, BMI, or sex. Bold indicates significance with *p* < 0.05.

## Data Availability

The datasets produced and examined in the present investigation can be accessed by the public through the NHANES website. (https://wwwn.cdc.gov/nchs/nhanes/Default.aspx, accessed on 18 March 2023).

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
