# Peer review of "Association of Dietary Vitamin C Consumption with Serum Klotho Concentrations"

_foods, 2023, doi:10.3390/foods12234230_

Round 1
Reviewer 1 Report
Comments and Suggestions for Authors
This review aims to investigate the association between vitamin C intake and serum Klotho levels based on data from the National Health and Nutrition Examination Survey (NHANES).
The objective of this article is clear. Minor revision is recommended.
In this study, vitamin C content in potato, legumes and pulses were excluded because they are not in the vegetable category, even though potato, legumes and pulses are high in vitamin C. For this reason, I recommend authors need to clearly write “dietary vitamin C from the consumption of fruits and vegetables” in the tile and texts through the manuscript.
In line 478, authors need to rewrite rather than “those” for clarification.
Comments on the Quality of English LanguageMinor editing of English language required
Author Response
Reviewer #1
We express our sincere gratitude to Reviewer #1 for offering a scholarly review and providing perceptive and detailed comments. These comments have proven to be of great importance and have served as a useful guideline for enhancing the quality of the manuscript. The following are detailed responses to each of the questions.
Comment 1: In this study, vitamin C content in potato, legumes and pulses were excluded because they are not in the vegetable category, even though potato, legumes and pulses are high in vitamin C. For this reason, I recommend authors need to clearly write “dietary vitamin C from the consumption of fruits and vegetables” in the tile and texts through the manuscript.
Response: We sincerely appreciate your invaluable advice. Actually, the data of dietary vitamin C consumption utilized in the main analysis of our manuscript was derived from all fruits and vegetables, with no exclusion of potatoes, legumes, or pulses. We sincerely apologize for any prior misunderstanding. To avoid potential misunderstanding for readers, we deleted the analysis that added vegetables and fruits as covariables additionally. Therefore, we did not add the state "dietary vitamin C from the consumption of fruits and vegetables" in the title and texts of the manuscript.
Comment 2: In line 478, authors need to rewrite rather than “those” for clarification.
Response: We appreciate your invaluable guidance. Based on your comments, we have carefully reviewed the entire manuscript and corrected any ambiguous words that might give rise to ambiguity. We hope our modification will be satisfactory to you. (Page 8, Line 286 and 304; Page 10, Line 418)
Reviewer 2 Report
Comments and Suggestions for Authors
Need better methods to explore the relationship.
Comments on the Quality of English LanguageLanguage needs some editing.
Author Response
Reviewer #2
We express our sincere gratitude to Reviewer #2 for offering a scholarly review and providing perceptive and detailed comments. These comments have proven to be of great importance and have served as a useful guideline for enhancing the quality of the manuscript. The following are detailed responses to each of the questions.
Comment 1: Need better methods to explore the relationship.
Response: A linear regression model was performed to explore the relationship between dietary vitamin C consumption and serum Klotho concentrations. Percentage changes was utilized to better explain the results of the generalized linear regression model. To explore the nonlinear and dose-response correlation between dietary vitamin C consumption and serum Klotho concentrations, stratified dietary vitamin C consumption and restricted cubic spline (RCS) analysis was conducted or further examination. These methodologies are well-established and scientifically sound for examining the relationship between continuous exposure and outcome variables, as supported by existing literature [1-3]. Consequently, we are confident in the appropriateness of the statistical approaches employed in this study.
Reference
- Tang, P.; Liao, Q.; Huang, H.; Chen, Q.; Liang, J.; Tang, Y.; Zhou, Y.; Zeng, X.; Qiu, X. Effects of urinary barium exposure on bone mineral density in general population. Environ Sci Pollut Res Int 2023, 30, 106038-106046, doi:10.1007/s11356-023-29791-0.
- Yan, W.; Sun, Y.; Wang, Y.; Liu, Y.; Yan, W.; Li, D.; Chen, M. The association of the serum levels of aldehydes with diabetes-related eye diseases: a cross-sectional population-based study. Environ Sci Pollut Res Int 2023, 30, 104713-104725, doi:10.1007/s11356-023-29750-9.
- Zhu, N.; Lin, S.; Yu, H.; Huang, W.; Cao, C. Association of Dietary Flavonoid Intake with Serum Cotinine Levels in the General Adult Population. Nutrients 2023, 15, doi:10.3390/nu15194126.
Reviewer 3 Report
Comments and Suggestions for Authors
The article is written in a way to includes journal rules and current issues. Tables and figures are expressed clearly and comprehensibly. I think it will contribute to the literature. I wish you did good work.
In order not to spoil the reading pleasure of the article, only the headings can be adjusted for alignment in PDF format articles. For example, on page 2, line 95, the title has shifted. Additionally, the font size should be reduced in Table 3. Because the expressions are very crowded and intertwined, making it difficult to analyze them in an understandable way.
Author Response
Reviewer #3
We express our sincere gratitude to Reviewer #3 for offering a scholarly review and providing perceptive and detailed comments. These comments have proven to be of great importance and have served as a useful guideline for enhancing the quality of the manuscript. The following are detailed responses to each of the questions.
Comment 1: In order not to spoil the reading pleasure of the article, only the headings can be adjusted for alignment in PDF format articles. For example, on page 2, line 95, the title has shifted. Additionally, the font size should be reduced in Table 3. Because the expressions are very crowded and intertwined, making it difficult to analyze them in an understandable way.
Response: We sincerely appreciate your invaluable advice. We have diligently reviewed and meticulously refined the article's formatting in accordance with your insightful suggestions. Consequently, we sincerely hope that you will find the modifications to your satisfaction.
Reviewer 4 Report
Comments and Suggestions for Authors
The article is a population-based study on the relationship between vitamin C consumption and levels of Klotho, a protein associated with anti-aging. The work is interesting, as it describes a new topic that assembles a contribution to the science of nutrition. Furthermore, it is original and presents an association unstudied relationship (vitamin C intake and serum Klotho levels). Despite this, I have some doubts about the study methodology. Furthermore, some flaws in the writing of the text need to be clarified.
Major points:
1) How did the authors consider nutrient-nutrient interactions when analyzing the level of vitamin C consumed? In this work, the 24-hour recall is used, which investigates all food intake in the last 24 hours. I imagine that the authors analyzed the foods and the amount consumed to assess the level of vitamin C consumed per day. However, the calculation of the level of vitamin C consumed also results from metabolic interactions that occur in different ways in organisms, such as nutrient-nutrient and drug-nutrient relationships. How can the authors confirm that these calculated data regarding vitamin C consumption are safe?
2) The study is carried out with a middle-aged and elderly population. I understand that the objective was to achieve a value of vitamin C consumed per day. However, would the most appropriate instrument for food anamnesis, considering the characteristics of the population studied, be the 24-hour recall? Do the authors know how it was applied and whether or not there was masking? What precautions were taken to confirm data security using this instrument?
3) The authors begin the results in line 394 stating that there is a significant association between vitamin C consumption and serum concentrations of Klotho. However, they end the paragraph disagreeing with this first statement, suggesting that their conclusion be cautiously viewed. This got confusing. The authors must rewrite the excerpt and clarify the result (significant statistical analysis) and the authors' hypothesis/opinion (belief in a positive result, but which was not significant), without leading to doubtful interpretations that could confuse the reader.
Other points:
4) At the beginning of page 11, the authors associate vitamin deficiency with the genesis of “diseases”: “Extensive epidemiological investigations have corroborated the widespread occurrence of vitamin C deficiency in the global population, leading to an increase in the incidence of diseases linked to such insufficiency [41]”. What are the diseases and what is the extensive research supporting this excerpt? The authors use only one reference.
5) In other parts of the manuscript, in addition to the one presented above, the authors use vague terms such as “many” and “several” and do not use references appropriately, leaving their arguments without a satisfactory scientific basis. Authors must review the entire manuscript and correct this problem.
6) On page 11, the authors attempt to infer causality between two distinct studies. They first cite the study [47], linking it to Klotho expression, and then describe the study [48], linking it to inflammation. Finally, they conclude that inflammation is related to the expression of Klotho. Based on this, did the study mentioned by [48] evaluate the relationship between inflammation and Klotho expression or did it just evaluate inflammation? It was unclear what the conclusion of the study [47] and study [48] was. If the answer to this question is “no”, the authors should rewrite the paragraph without drawing unbiased conclusions.
Author Response
Reviewer #4
We express our sincere gratitude to Reviewer #4 for offering a scholarly review and providing perceptive and detailed comments. These comments have proven to be of great importance and have served as a useful guideline for enhancing the quality of the manuscript. The following are detailed responses to each of the questions.
Comment 1: How did the authors consider nutrient-nutrient interactions when analyzing the level of vitamin C consumed? In this work, the 24-hour recall is used, which investigates all food intake in the last 24 hours. I imagine that the authors analyzed the foods and the amount consumed to assess the level of vitamin C consumed per day. However, the calculation of the level of vitamin C consumed also results from metabolic interactions that occur in different ways in organisms, such as nutrient-nutrient and drug-nutrient relationships. How can the authors confirm that these calculated data regarding vitamin C consumption are safe?
Response: The calculations of nutrients intake in NHANES were acquired with the assistance of the USDA's (United States Department of Agriculture) dietary data collection tool—Automated Multiple Pass Method (AMPM). AMPM upholds data accuracy and uniformity through distinct interview stages and quality control measures. The 24-hour dietary review in NHANES did not account for nutrient-nutrient interactions, which may be attributed to the complexity of the types of nutrients and the unclear composition of nutrients in the diet, rendering it challenging to incorporate nutrient-nutrient interactions in the dietary review. In large-scale epidemiological studies, particularly in nutritional epidemiology, which investigates the link between dietary nutrients and human health, the accuracy of assessing nutrient intake is influenced by various factors, including human digestive and absorption capacity, nutrient-nutrient interactions, nutrient-drug interactions, and genetic metabolism in vivo. We concur with the reviewer's observation that nutrient-nutrient interactions and nutrient-drug interactions can impact the evaluation of dietary vitamin C intake. However, it is essential to acknowledge that considering interactions among numerous nutrients remains challenging due to the limitations of current assessment methods and the complex web of interactions within a wide array of nutrients. The absence of a comprehensive examination of nutrient-nutrient and nutrient-drug interactions is one of the limitations of this study, which will be addressed in future research. In the limitations section, we have included the following statement: "The assessment of dietary vitamin C consumption may be influenced by factors such as an individual's nutritional status, intestinal health, nutrient-nutrient interactions, nutrient-drug interactions, and other variables that could affect the evaluation of dietary vitamin C consumption" (Page 10, Lines 410-413). Furthermore, as NHANES only provided 24-hour recalls for determining dietary vitamin C content from 2007 to 2016, we concur with the idea that serum vitamin C data serve as a more robust indicator of vitamin C intake. Regrettably, due to unavailable of present data, we were unable to select a more compelling serum vitamin C dataset for analysis. This limitation has been added (Page10, Line 408-410).
To assess the reliability of dietary data, we meticulously screened for inaccuracies in result reporting during our analysis. The reliability of the serial number (DR1DRSTZ - Dietary recall status) can be verified at the following URL: https://wwwn.cdc.gov/Nchs/Nhanes/2015-2016/DR1TOT_I.htm#DR1DRSTZ. In our current analysis, the average dietary vitamin C intake was 81mg/day, which closely aligns with previous reports involving similar populations [1]. This indirectly underscores the reliability of dietary vitamin C intake assessment in this study. In response to reviewers' concerns regarding the impact of nutrient-nutrient interactions on serum Klotho levels, we identified several nutrients, based on previous studies, that have been demonstrated to interact with both dietary vitamin C and serum Klotho levels. An analysis of the interaction between supplementation and dietary vitamin C on serum Klotho concentration (refer to Table R1) was conducted, including vitamin A, vitamin B12, vitamin D, vitamin E, dietary fiber, carbohydrate, total folic acid, and copper. The results indicated interactions between vitamin A [2], vitamin B12 [3], vitamin D [4], vitamin E [5], dietary fiber [6], carbohydrate [7], total folic acid [8], and copper [9] with both dietary vitamin C and serum Klotho concentration. Since the primary focus of this study was not nutrient-nutrient interactions, this analysis was not included in the supplementary materials.
Table R1. Relationship between nutrient-nutrient interactions with dietary vitamin C consumption and serum Klotho concentrations
|
Nutrient-nutrient interactions |
*P for interaction |
|
Vitamin A |
0.022 |
|
Vitamin B12 |
0.595 |
|
Vitamin D |
0.730 |
|
Vitamin E |
0.532 |
|
Dietary fiber |
0.026 |
|
Carbohydrate |
0.948 |
|
Total folate |
0.044 |
|
Copper |
<0.001 |
Note: *P value for interaction of vitamin C intake with vitamin A、vitamin B12、vitamin D、vitamin E、dietary fiber、carbohydrate、total folate or copper. Bold indicates the significant with P < 0.05.
Comment 2: The study is carried out with a middle-aged and elderly population. I understand that the objective was to achieve a value of vitamin C consumed per day. However, would the most appropriate instrument for food anamnesis, considering the characteristics of the population studied, be the 24-hour recall? Do the authors know how it was applied and whether or not there was masking? What precautions were taken to confirm data security using this instrument?
Response: We concur with you that dietary assessments among middle-aged and elderly individuals might be susceptible to bias stemming from memory or cognitive decline. In the target demographic of this study (aged 40-79), the predominant method employed for dietary assessment is the 24-hour dietary recall method compared to food frequency questionnaires (FFQ) [10]. To our knowledge, a more accurate method might be to keep a portion of the food for testing data and then conduct a dietary review. Nonetheless, this method poses greater challenges when applied to large-scale nutritional epidemiology studies.
The 24-hour dietary recall in NHANES, a commonplace method in nutritional epidemiology, follows a general operating content: Participants are given three-dimensional models (measuring cups and spoons, a ruler, and two household spoons) and/or USDA's Food Model Booklet (containing drawings of various sizes of glasses, mugs, bowls, mounds, circles, and other measures) to estimate food amounts. Individual foods data contains one record per food for each survey participant. Foods are identified by USDA food codes. Each record contains information about when and where the food was consumed, whether the food was eaten in combination with other foods, amount eaten, and amounts of nutrients provided by the food. Total nutrient intakes data contains one record per day for each survey participant. Each record contains daily totals of food energy and nutrient intakes, daily intake of water, intake day of week, total number foods reported, and whether intake was usual, much more than usual or much less than usual. The Day 1 file also includes salt use in cooking and at the table; whether on a diet to lose weight or for other health-related reason and type of diet; and frequency of fish and shellfish consumption.
Masking phenomena are present in dietary assessments. Even with the provision of food models, variations in dietary data can be influenced by individual cognitive disparities or discrepancies in memory. Furthermore, given that NHANES solely offered the 24-hour recall method for ascertaining dietary vitamin C content from 2007 to 2016, we concur that utilizing serum vitamin C data is a more suitable approach for analysis. Nonetheless, owing to unavailable data, we regretfully lack the means to choose a more robust serum vitamin C data source for analysis. This represents one of the article's limitations, which we have added in the discussion (Page 11, Line 408-410).
The data utilized in this study is reliable, and any unreliable data is excluded from analysis. NHANES uses the following precautions to ensure data security: ①Training and Certification: Ensure that investigators receive specialized training for the accurate collection of dietary information. ②Utilize Standardized Tools: Employ standardized dietary survey instruments, such as food intake questionnaires, to ensure uniformity and comparability. ③Clear records: Make sure records are clear, detailed, and accurate, including food types, portion sizes, etc. ④Food models: Food models or pictures are provided to help participants better estimate their food intake. ⑤Pay attention to details: Details are crucial, including the specific description of the food. ⑥Time logging: Ensuring when and where participants provided food to more accurately assess their eating habits. ⑦Quality control: Implement quality control measures, including data inspection, calibration and verification, to ensure data consistency and accuracy. ⑧Data analysis: It is recommended to analyze the data using appropriate statistical methods to explore the relationship between food intake and the research question.
Comment 3: The authors begin the results in line 394 stating that there is a significant association between vitamin C consumption and serum concentrations of Klotho. However, they end the paragraph disagreeing with this first statement, suggesting that their conclusion be cautiously viewed. This got confusing. The authors must rewrite the excerpt and clarify the result (significant statistical analysis) and the authors' hypothesis/opinion (belief in a positive result, but which was not significant), without leading to doubtful interpretations that could confuse the reader.
Response: Thank you for your valuable advice. To make the statement more clear, we have amended it accordingly (Page 10, Line 346-350 and 361-364): “... In this study, to explore the potential age-specific relationship between dietary vitamin C consumption and serum Klotho concentrations, we additionally performed a subgroup analysis. Stratified analysis results showed that increased dietary vitamin C consumption was significantly associated with increased serum Klotho concentrations in older participants, but not in middle-aged participants. ...Although the vitamin C-Klotho relationship was more prominent among elderly adults, the age-vitamin C interaction displayed insignificant modification effect on the levels of Klotho. Therefore, age-specific relationship of dietary vitamin C consumption with serum Klotho concentrations should be viewed with caution.”
Comment 4: At the beginning of page 11, the authors associate vitamin deficiency with the genesis of “diseases”: “Extensive epidemiological investigations have corroborated the widespread occurrence of vitamin C deficiency in the global population, leading to an increase in the incidence of diseases linked to such insufficiency [41]”. What are the diseases and what is the extensive research supporting this excerpt? The authors use only one reference.
Response: According to your suggestion, we have revised the sentence as follows:
“Extensive epidemiological investigations have corroborated the widespread occurrence of vitamin C deficiency in the global populace, leading to a heightened incidence of diseases linked to such insufficiency, which can include potentially fatal conditions like scurvy. This deficiency has been linked to various health issues, including cardiovascular disease, conges-tive heart failure, malignancies, chronic inflammation, metabolic disorders, and non-communicable diseases such as cataracts [33,41,44,45].” (Page 9, Line 314-319)
Comment 5: In other parts of the manuscript, in addition to the one presented above, the authors use vague terms such as “many” and “several” and do not use references appropriately, leaving their arguments without a satisfactory scientific basis. Authors must review the entire manuscript and correct this problem.
Response: Thank you for your valuable advice. Based on your comments, we have carefully reviewed the entire manuscript and added corresponding references. See the following locations in the manuscript for details (Page 10, Line 379).
Comment 6: On page 11, the authors attempt to infer causality between two distinct studies. They first cite the study [47], linking it to Klotho expression, and then describe the study [48], linking it to inflammation. Finally, they conclude that inflammation is related to the expression of Klotho. Based on this, did the study mentioned by [48] evaluate the relationship between inflammation and Klotho expression or did it just evaluate inflammation? It was unclear what the conclusion of the study [47] and study [48] was. If the answer to this question is “no”, the authors should rewrite the paragraph without drawing unbiased conclusions.
Response: According to your suggestion, we have revised the sentence as follows:
“On the other hand, both systemic and local inflammation may reduce the expression level of Klotho protein in the kidney [47]. In animal experiments as well as clinical trials, vitamin C has been shown to alleviate the inflammatory response by modulating inflammatory factor production as well as inhibiting inflammatory mediator infiltration [48]. In addition, the positive regulatory effect of vitamin C by inhibiting oxidative stress levels and thereby increasing Klotho levels has been demonstrated in rats with a hyperoxaluria model [18]. Therefore, we speculate that vitamin C supplementation may help to combat the down-regulation of Klotho protein expression levels caused by inflammation. However, in the population, the correlation between vitamin C and Klotho and the specific regulatory mechanisms have not been reported.” (Page 9, Line 335-344)
Reference
- U.S. DEPARTMENT OF AGRICULTURE. What We Eat In America. Available online: https://www.ars.usda.gov/nea/bhnrc/fsrg.
- Rausch, S.; Barholz, M.; Föller, M.; Feger, M. Vitamin A regulates fibroblast growth factor 23 (FGF23). Nutrition 2020, 79-80, 110988, doi:10.1016/j.nut.2020.110988.
- Choi, J.-Y.; Min, J.-Y.; Min, K.-B. Anti-aging protein klotho was associated with vitamin B12 concentration in adults. Medicine (Baltimore) 2022, 101, e30710, doi:10.1097/MD.0000000000030710.
- Tuohimaa, P. Vitamin D and aging. J Steroid Biochem Mol Biol 2009, 114, 78-84.
- Jaturakan, O.; Buranakarl, C.; Dissayabutra, T.; Chaiyabutr, N.; Kijtawornrat, A.; Rungsipipat, A. Changes of Klotho protein and Klotho mRNA expression in a hydroxy-L-proline induced hyperoxaluric rat model. J Vet Med Sci 2017, 79, 1861-1869, doi:10.1292/jvms.17-0340.
- Liu, S.; Wu, M.; Wang, Y.; Xiang, L.; Luo, G.; Lin, Q.; Xiao, L. The Association between Dietary Fiber Intake and Serum Klotho Levels in Americans: A Cross-Sectional Study from the National Health and Nutrition Examination Survey. Nutrients 2023, 15, doi:10.3390/nu15143147.
- Xiang, L.; Wu, M.; Wang, Y.; Liu, S.; Lin, Q.; Luo, G.; Xiao, L. Inverse J-Shaped Relationship of Dietary Carbohydrate Intake with Serum Klotho in NHANES 2007-2016. Nutrients 2023, 15, doi:10.3390/nu15183956.
- Lucock, M.; Yates, Z.; Boyd, L.; Naylor, C.; Choi, J.-H.; Ng, X.; Skinner, V.; Wai, R.; Kho, J.; Tang, S.; et al. Vitamin C-related nutrient-nutrient and nutrient-gene interactions that modify folate status. Eur J Nutr 2013, 52, 569-582, doi:10.1007/s00394-012-0359-8.
- Ostojic, S.M.; Hillesund, E.R.; Øverby, N.C.; Vik, F.N.; Medin, A.C. Individual nutrients and serum klotho levels in adults aged 40-79 years. Food Sci Nutr 2023, 11, 3279-3286, doi:10.1002/fsn3.3310.
- Dietary Assessment Primer. NATIONAL CANCER INSTITUTE. Available online: https://dietassessmentprimer.cancer.gov/approach/principles.html